## PERSPECTIVE

# *In utero* electronic cigarette exposure may have mind blowing impacts on the offspring

Sarah Commodore [ID],
Brook Gebremariam
and Chukwudike Igwe [ID]

*Department of Environmental and Occupational Health, Indiana University, Bloomington, IN, USA*

Email: scommod@iu.edu

Handling Editors: Laura Bennet & Justin Dean

The peer review history is available in the Supporting information section of this article (https://doi.org/10.1113/JP290093#support-information-section).

We commend Mills and colleagues (Mills et al., 2025) for their salient study on the effects of maternal electronic cigarette (e-cig) use during pregnancy, on cerebrovascular and neurocognitive health of offspring during their adult lives both in the presence and absence of nicotine. They tested the hypothesis that maternal vaping in a rat model would adversely affect cerebrovascular functions. These effects would be independent of nicotine and observed at different wattages. After confirming pregnancy, the authors randomly assigned Sprague–Dawley rat dams to five exposure categories: (a) e-cig with 0 mg/mL nicotine at 5 W, (b) e-cig with 0 mg/mL nicotine at 30 W, (c) e-cig with 50 mg/mL nicotine at 5 W, (d) e-cig with 50 mg/mL nicotine at 30 W, and (e) ambient air, which served as the control group. The 5 W and 30 W options were selected for human relevance as the former is similar to the power level operated by pod-style device users, while the latter would be for experienced users operating mod-tank styles. The team observed that long-term cerebrovascular dysfunction persisted throughout the lifetime of an *in utero* exposed offspring and that the higher wattage was associated with worse endothelial cell dysfunction, while the presence of nicotine was associated with cognitive impairment. Lastly, e-cig exposures *in utero* were associated with the differential gene and protein expression that can subsequently lead to accelerated cellular senescence and the risk for neuro-degenerative diseases.

This study marks a leap forward in the understanding of how nicotine may contribute to a wide range of neuro-degenerative disorders. Although nicotine may not always be associated with the aetiology of cerebrovascular impairment, its presence contributed to neurocognitive deficits and the severity of neuronal damage in the aforementioned study. Offspring who experienced e-cig exposure during pregnancy, regardless of the wattage or nicotine, had decreased *SIRT1*, elevated *NOX1*, and displayed pathology that is associated with Alzheimer's Disease. Indeed, the study highlights opportunities, particularly for those interested in tobacco cessation, to advocate for policies, and specific public health actions that can help protect the sensitive fetal brain from the effects of vaping during pregnancy.

First, we have the opportunity to act now, even as we await the results of future research and clinical studies. The long-term e-cig health impacts in humans remain unknown, and most studies on vape product use are in adult populations, some of whom have comorbidities. However, currently there are no studies with decades-old results because e-cig product use research is novel (<20 years) and quickly evolving. While further studies are needed to understand the neurological health impacts of long-term vape products on human users, as well as their offspring, we can continue to advocate for vaping cessation during pregnancy. There is no reason to wait for the future results before taking a stand; we can err on the side of caution and focus on helping pregnant women quit vaping.

Secondly, cerebrovascular impairment in offspring with maternal e-cig exposure appears to be dependent on the wattage of the e-cig device and not the presence of nicotine. Variable power settings, along with multiple flavours may introduce a new keg of toxicity previously not encountered in traditional tobacco research. The solutions in e-cigs, and their resulting aerosols (in the presence or absence of nicotine), may contain carcinogens, as well as organic and inorganic constituents, to which users and non-users in close proximity can be exposed (Walley & Jenssen, 2015). Given the attractive flavours, marketing, design, and its appeal, e-cigs have the potential to reverse decades of progress achieved in nicotine and tobacco product use (Walley et al., 2019). We could keep the momentum going and talk about device settings, characteristics, and user behaviours with e-cig users to reduce adverse health impacts on themselves as well as their offspring.

Lastly, e-cig users may give birth to future users, and there is an occasion to break this cycle. An average of 8.7 million children aged 17 or younger live in US households with at least one parent who had a substance use disorder (Lipari & Van Horn, 2017). Such statistics highlight the potential breadth of substance use prevention and treatment needs for the whole family from substance use treatment for affected adults to prevention and supportive services for children. Furthermore, the literature shows that children whose parents smoke are more likely to become smokers (Leonardi-Bee et al., 2011), and this may be similar for e-cig use. Mills and colleagues showed that rodent offspring who had *in utero* exposure to e-cigs exhibited long-lasting cerebrovascular and neurocognitive dysfunction in adulthood, indicating that vaping during pregnancy may not be innocuous. If parents, particularly mothers, understand the health impacts of vaping from conception to birth and the potential for passive and environmental exposures, we are a step closer to intervening and lessening the health impact of e-cig exposures.

In sum, maternal vaping during pregnancy may exert long-lasting adverse effects on the developing brain of offspring, beginning from the womb, and these effects may persist into adulthood. The plausible role of nicotine and the infinite range of flavours and user settings may throw an additional curveball into the field of tobacco research. For clinicians, policymakers and other stakeholders, the ultimate goal would be to advocate for a vape-free pregnancy, since this would allow for ideal brain and overall development of the offspring across the entire lifespan.

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

## Additional information

### Competing interests

None declared.

### Author contributions

S.C., B.G. and C.I. drafted the work. S.C., B.G. and C.I. revised it critically for important intellectual content. S.C. acquired the funding. All authors have approved the final version of the manuscript and agreed to be accountable for all aspects of the work. All persons designated as authors qualify for authorship, and all those who qualify for authorship are listed.

### Funding

This work was supported in part by the Indiana University School of Public Health-Bloomington and NIH/NIEHS (1R01ES035694-01A1).

## Keywords

brain, development, electronic cigarettes, *in utero* exposures, vaping

## Supporting information

Additional supporting information can be found online in the Supporting Information section at the end of the HTML view of the article. Supporting information files available:

**Peer Review History**

