## [Peer Review History · The Journal of Physiology]

***In utero* electronic cigarette exposure may have mind blowing impacts on the offspring**

Sarah Commodore, Brook Gebremariam, and Chukwudike Igwe
DOI: 10.1113/JP290093

Corresponding author(s): Sarah Commodore (scommod@iu.edu)

Review Timeline:

Submission Date:

19-Sep-2025

Accepted:

25-Sep-2025

Senior Editor: Laura Bennet

Reviewing Editor: Justin Dean

Transaction Report:

Dear Dr Commodore,

Re: JP-P-2025-290093 "***In utero* electronic cigarette exposure may have mind blowing impacts on the offspring**" by Sarah Commodore, Brook Gebremariam, and Chukwudike Igwe

We are pleased to tell you that your paper has been accepted for publication in The Journal of Physiology.

Yours sincerely,

Laura Bennet
Senior Editor
The Journal of Physiology

If you would like to receive our 'Research Roundup', a monthly newsletter highlighting the cutting-edge research published in The Physiological Society's family of journals (The Journal of Physiology, Experimental Physiology, Physiological Reports, The Journal of Nutritional Physiology, and The Journal of Precision Medicine: Health and Disease), please click this link, fill in your name and email address and select 'Research Roundup':

<https://www.physoc.org/journals-and-media/membernews>

- You can help your research get the attention it deserves! Check out Wiley's free Promotion Guide for best-practice recommendations for promoting your work at: www.wileyauthors.com/eeo/guide. You can learn more about Wiley Editing Services which offers professional video, design, and writing services to create shareable video abstracts, infographics, conference posters, lay summaries, and research news stories for your research at: www.wileyauthors.com/eeo/promotion.

The Corresponding Author will receive an email from Wiley with details on how to register or log-in to Wiley Authors Services where you will be able to place an order

EDITOR COMMENTS

Reviewing Editor:

Thank you for your submission. The reviewer has reviewed your manuscript, and is happy with the perspective article as written.

REFEREE COMMENTS

Referee #1:

This is a perspective article based on Mills et al paper. The authors have provided a concise and clear (and even witty) narrative of the work presented. The perspective takes a balanced position and helps to deliver an overall message that vaping during pregnancy is unlikely to be safe. I have no changes to suggest. Well done!